# ADAPTIVE QUANTIZATION OF NEURAL NETWORKS

**Soroosh Khoram**
Department of Electrical and Computer Engineering
University of Wisconsin - Madison
`khoram@wisc.edu`

**Jing Li**
Department of Electrical and Computer Engineering
University of Wisconsin - Madison
`jli@ece.wisc.edu`

## ABSTRACT

Despite the state-of-the-art accuracy of Deep Neural Networks (DNN) in various classification problems, their deployment onto resource constrained edge computing devices remains challenging due to their large size and complexity. Several recent studies have reported remarkable results in reducing this complexity through quantization of DNN models. However, these studies usually do not consider the changes in the loss function when performing quantization, nor do they take the different importances of DNN model parameters to the accuracy into account. We address these issues in this paper by proposing a new method, called adaptive quantization, which simplifies a trained DNN model by finding a unique, optimal precision for each network parameter such that the increase in loss is minimized. The optimization problem at the core of this method iteratively uses the loss function gradient to determine an error margin for each parameter and assigns it a precision accordingly. Since this problem uses linear functions, it is computationally cheap and, as we will show, has a closed-form approximate solution. Experiments on MNIST, CIFAR, and SVHN datasets showed that the proposed method can achieve near or better than state-of-the-art reduction in model size with similar error rates. Furthermore, it can achieve compressions close to floating-point model compression methods without loss of accuracy.

## 1 INTRODUCTION

Deep Neural Networks (DNNs) have achieved incredible accuracies in applications ranging from computer vision (Simonyan & Zisserman, 2014) to speech recognition (Hinton et al., 2012) and natural language processing (Devlin et al., 2014). One of the key enablers of the unprecedented success of DNNs is the availability of *very large model sizes*. While the increase in model size improves the classification accuracy, it inevitably increases the computational complexity and memory requirement needed to train and store the network. This poses challenges in deploying these large models in resource-constrained edge computing environments, such as mobile devices. These challenges motivate *neural network compression*, which exploits the redundancy of neural networks to achieve drastic reductions in model sizes. The state-of-the-art neural network compression techniques include weight quantization (Courbariaux et al., 2015), weight pruning (Han et al., 2015), weight sharing (Han et al., 2015), and low rank approximation (Zhao et al., 2017). For instance, weight quantization has previously shown good accuracy with fixed-point 16-bit and 8-bit precisions (Suda et al., 2016; Qiu et al., 2016). Recent works attempt to push that even further towards reduced precision and have trained models with 4-bit, 2-bit, and 1-bit parameters using quantized training methods (Hubara et al., 2016; Zhou et al., 2016; Courbariaux & Bengio, 2016; Courbariaux et al., 2015).

Although these quantization methods can significantly reduce model complexity, they generally have two key constraints. First, they ignore the accuracy degradation resulting from quantization, during the quantization, and tend to remedy it, separately, through quantized learning schemes. However, such schemes have the disadvantage of converging very slowly compared to full-precision learning methods. Second, they treat all network parameters similarly and assign them the same *quantization width*[1]. This is while previous works (Courbariaux & Bengio, 2016; Hubara et al., 2016; Han et al., 2015) have shown different parameters do not contribute to the model accuracy equally. Disregarding this variation limits the maximum achievable compression.

---

[1] Number of bits used to store the fixed-point quantized value.

In this paper, we address the aforementioned issues by proposing *adaptive quantization*. To take the different importances of network parameters into account, this method quantizes each network parameter of a trained network by a unique quantization width. This way, parameters that impact the accuracy the most can be represented using higher precisions (larger quantization widths), while low-impact parameters are represented with fewer bits or are pruned. Consequently, our method can reduce the model size significantly while maintaining a certain accuracy. The proposed method monitors the accuracy by incorporating the loss function into an optimization problem to minimize the models. The output of the optimization problem is an error margin associated to each parameter. This margin is computed based on the loss function gradient of the parameter and is used to determine its precision. We will show that the proposed optimization problem has a closed-form approximate solution, which can be iteratively applied to the same network to minimize its size. We test the proposed method using three classification benchmarks comprising MNIST, CIFAR-10, and SVHN. We show that, across all these benchmarks, we can achieve near or better compressions compared to state-of-the-art quantization techniques. Furthermore, we can achieve compressions similar to the state-of-the-art pruning and weight-sharing techniques which inherently require more computational resources for inference.

## 2 PROPOSED QUANTIZATION ALGORITHM

Despite their remarkable classification accuracies, large DNNs assimilate redundancies. Several recent works have studied these redundancies by abstracting the network from different levels and searching for, in particular, redundant filters (DenseNets [Huang et al., 2016]) and redundant connections (Deep Compression [Han et al., 2015]). In this work, we present an even more fine-grained study of redundancy and extend it to fixed-point quantization of network parameters. That is, we approximate the minimum size of the network when each parameter is allowed to have a distinct number of precision bits. Our goal here is to represent parameters with high precisions only when they are critical to the accuracy of the network. In this sense, our approach is similar to weight pruning (Han et al., 2015) which eliminates all but the essential parameters, producing a sparse network. In the rest of this section, we first formally define the problem of minimizing the network size as an optimization problem. Then, we propose a trust region technique to approximately solve this problem. We will show that, in each iteration of the trust region method, our approach has a straightforward, closed-form solution. Finally, we will explain how the hyper parameters in the algorithm are chosen and discuss the implications of the proposed technique.

It is also important to discuss the benefits of this fine-grained quantization for performance. In particular, besides its storage advantages, we argue that this method can reduce the computation, if the target hardware can take advantage of the non-standard, yet small quantization depths that our approach produces. We note that there exist techniques on CPU and GPU for fixed-point arithmetics with non-standard quantization widths (e.g. SWAR [Cameron & Lin, 2009]). But, we believe our proposed quantization is ideal for FPGAs and specialized hardware. The flexibility of these platforms allows for design of efficient computation units that directly process the resulting fixed-point quantized parameters. This was recently explored by Albericio et al. (2017), who designed specialized computation units for variable bit-width parameters that eliminated ineffectual computations. By minimizing the overall number of bits that need to be processed for a network, the proposed quantization achieves the same effect.

### 2.1 PROBLEM DEFINITION

A formal definition of the optimization problem that was discussed previously is presented here. Then, key characteristics of the objective function are derived. We will use these characteristics later when we solve the optimization problem.

We minimize the aggregate bit-widths of all network parameters while monitoring the training loss function. Due to the quantization noise, this function deviates from its optimum which was achieved through training. This noise effect can be controlled by introducing an upper bound on the loss function in order to maintain it reasonably close to the optimum. Consequently, the solution of this minimization problem represents critical parameters with high precision to sustain high accuracy and

assigns low precisions to ineffectual ones or prunes them. The problem described here can be defined in formal terms as below.

$$\min_{W} N_Q(W) = \sum_{i=1}^{n} N_q(\omega_i) \qquad (1)$$

$$\mathcal{L}(W) \leq \bar{\ell} \qquad (2)$$

Here, $n$ is the number of model parameters, $W = [\omega_1 ... \omega_n]^T$ is a vector of all model parameters, $\omega_i$ is the $i$-th model parameter, and the function $N_q(\omega_i)$ is the minimum number of bits required to represent $\omega_i$ in its fixed-point format. As a result, $N_Q(W)$ is the total number of bits required to represent the model. In addition, $\mathcal{L}(W)$ is the loss function value of the model $W$ over the training set (or a minibatch of the training set). Finally, $\bar{\ell}$ ($\geq \mathcal{L}(W_0)$) is a constant upper bound on the loss of the neural network, which is used to bound the accuracy loss, and $W_0$ is the vector of initial model parameters before quantization.

The optimization problem presented in Equations 1 and 2 is difficult to solve because of its non-smooth objective function. However, a smooth upper limit can be found for it. In the lemma below we derive such a bound. To our knowledge, this is the first time such a bound has been developed in the literature.

**Lemma 2.1.** *Given a vector of network parameters $W$, a vector of tolerance values $T = [\tau_1 ... \tau_n]^T$, and a vector of quantized network parameters $W_q = [\omega_1^q ... \omega_n^q]^T$, such that each $\omega_i^q, i \in [1, n]$ has a quantization error of at most $\tau_i$, meaning it solves the constrained optimization function:*

$$\min_{\omega^q} N_q(\omega_i^q) \qquad (3)$$

$$|\omega_i^q - \omega_i| \leq \tau_i \qquad (4)$$

*We have that:*

$$N_Q(W) \leq \Phi(T) \qquad (5)$$

*Where $\Phi(T)$ is a smooth function, defined as:*

$$\Phi(T) = -\sum_{i=1}^{n} \log_2 \tau_i \qquad (6)$$

*Proof.* In Equation 3, we allow each parameter $\omega_i$ to be perturbed by at most $\tau_i$ to generate the quantized parameter $\omega_i^q$. The problem of Equation 3 can be easily solved using Algorithm 1, which simply checks all possible solutions one-by-one. Here, we allocate at most 32 bits to each parameter. This should be more than enough as previous works have demonstrated that typical DNNs can be easily quantized to even 8 bits without a significant loss of accuracy.

---

**Algorithm 1** Quantization of a parameter

---

**procedure** QUANTIZE_PARAMETER($\omega_i, \tau_i$)
    $\omega_i^q \leftarrow 0$
    $N_q^i \leftarrow 0$
    **while** $N_q^i \leq 32$ **do**
        **if** $|\omega_i^q - \omega_i| \leq \tau_i$ **then**
            $break$
        **end if**
        $\omega_i^q \leftarrow round(2^{N_q^i} \omega_i)/2^{N_q^i}$
        $N_q^i \leftarrow N_q^i + 1$
    **end while**
    **return** $N_q^i, \omega_i^q$
**end procedure**

---

Algorithm 1 estimates $\omega_i$ with a rounding error of up to $\frac{1}{2^{N_q^i+1}}$. Thus, for the worst case to be consistent with the constraint of Equation 3, we need:

$$\frac{1}{2^{N_q(\omega_i)+1}} \leq \tau_i \Rightarrow N_q(\omega_i) + 1 \geq -\log_2 \tau_i \tag{7}$$

In this case the minimization guarantees that the smallest $N_q(\omega_i)$ is chosen. This entails:

$$N_q(\omega_i) \leq -\lceil \log_2 \tau_i \rceil - 1 \leq -\log_2 \tau_i \tag{8}$$

In general, we can expect $N_q(\omega_i)$ to be smaller than this worst case. Consequently:

$$N_Q(W) = \sum_{i=1}^{n} N_q(\omega_i) \leq -\sum_{i=1}^{n} \log_2 \tau_i = \Phi(T) \tag{9}$$

$$\square$$

We should note that for simplicity Algorithm 1 assumes that parameters $\omega_i$ are unsigned and in the range $[0, 1]$. In practice, for signed values, we quantize the absolute value using the same method, and use one bit to represent the sign. For models with parameters in a range larger than $[-1, 1]$, say $[-r, r]$ with $r > 1$, we perform the quantization similarly. The only difference is that we first scale the parameters and their tolerances by $\frac{1}{r}$ to bring them to $[-1, 1]$ range and then quantize the parameters. When computing the output of a quantized layer, then, we multiply the result by $r$ to account for the scale. Following this, the same equations as above will hold.

## 2.2 SOLVING THE OPTIMIZATION PROBLEM

The optimization problem of Equations 1 and 2 presents two key challenges. First, as was mentioned previously, the objective function is non-smooth. Second, not only is the constraint (Equation 2) non-linear, but also it is unknown. In this section, we present a method that circumvents these challenges and provides an approximate solution for this problem.

We sidestep the first challenge by using the upper bound of the objective function, $\Phi(T)$. Particularly, we approximate the minimum of $N_Q(W)$ by first finding the optimal $T$ for $\Phi(T)$ and then calculating the quantized parameters using Algorithm 1. This optimization problem can be defined as: minimize $\Phi(T)$ such that if each $\omega_i$ is perturbed by, at most, $\tau_i$, the loss constraint of Equation 2 is not violated:

$$\min_T \Phi(T) \tag{10}$$

$$\mathcal{L}(W_0 + \Delta W) \leq \bar{\ell} \tag{11}$$

$$\forall \Delta W = [\Delta \omega_1 ... \Delta \omega_n]^T \quad such\ that \quad \forall i \in [1, n] : |\Delta \omega_i| \leq \tau_i \tag{12}$$

It is important to note that although the upper bound $\Phi(T)$ is smooth over all its domain, it can be tight only when $\forall i \in [1, n] : \tau_i = 2^{-k}, k \in \mathbb{N}$. This difference between the objective function and $\Phi(T)$ means that it is possible to iteratively reduce $N_Q(W)$ by repeating the steps of the indirect method, described above, and improve our approximation of the optimal quantization.

Such an iterative algorithm would also help address the second challenge. Specifically, it allows us to use a simple model of the loss function (e.g. a linear or quadratic model) as a stand-in for our complex loss function. If in some iteration of the algorithm the model is inaccurate, it can be adjusted in the following iteration. In our optimization algorithm, we will use a linear bound of the loss function and adopt a *Trust Region* method to monitor its accuracy.

Trust region optimization methods iteratively optimize an approximation of the objective function. In order to guarantee the accuracy of this approximation, they further define a neighborhood around each point in the domain, called a trust region, inside which the model is a "sufficiently good" approximation of the objective function. The quality of the approximation is evaluated each iteration and the size of the trust region is updated accordingly. In our algorithm we adopt a similar technique, but apply it to the constraint instead of the objective function.

**Lemma 2.2.** *We refer to $m_\ell : \mathcal{R}^n \to \mathcal{R}$ as an accurate estimation of $\mathcal{L}(W)$ in a subdomain of $\mathcal{L}$ like $\mathcal{D} \subset \mathcal{R}^n$, if:*

$$\forall W \in \mathcal{D} : m_\ell(W) \leq \bar{\ell} \Rightarrow \mathcal{L}(W) \leq \bar{\ell} \tag{13}$$

*Now, let the radius $\Delta$ specify a spherical trust region around the point $W_0$ where the loss function $\mathcal{L}$ is accurately estimated by its first-order Taylor series expansion. Then, the constraint of Equation 16, when $\|T\|_2 \leq \Delta$, is equivalent to:*

$$m_\ell(T) = \mathcal{L}(W_0) + G^T T \leq \bar{\ell} \tag{14}$$

*Where $G = [g_1...g_n]^T$ and $g_i = |[\nabla_W \mathcal{L}(W_0)]_i|, \forall i \in [1, n]$.*

*Proof.* Since $\|T\|_2 \leq \Delta$, then for $\Delta W$ as defined in Equation 12, $W_0 + \Delta W \in \mathcal{D}$. Therefore, we can write:

$$\mathcal{L}(W_0) + \nabla_W \mathcal{L}(W_0)^T \Delta W \leq \mathcal{L}(W_0) + G^T T = m_\ell(T) \leq \bar{\ell} \Rightarrow \mathcal{L}(W_0 + \Delta W) \leq \bar{\ell} \tag{15}$$

$\square$

We note that we did not make any assumptions regarding the point $W_0$ and the result can be extended to any point in $\mathcal{R}^n$. Consequently, if we define such a trust region, we can simply use the following problem as a subproblem that we repeatedly solve in successive iterations. We will present the solution for this subproblem in the next section.

$$\min_T \Phi(T) \tag{16}$$

$$m_\ell(T) \leq \bar{\ell} \tag{17}$$

Algorithm 2 summarizes the proposed method of solving the original optimization problem (Equation 1). This algorithm first initializes the initial trust region radius $\Delta_0$ and the loss function bound $\bar{\ell}$. It also quantizes the input floating-point parameters $W_0$ with 32 bits to generate the initial quantized parameters $W^0$. Here the function *Quantize(.,.)* refers to a pass of Algorithm 1 over all parameters. Thus, using 0 tolerance results in quantization with 32 bits.

Subsequently, Algorithm 2 iteratively solves the subproblem of Equation 16 and calculates the quantized parameters. In iteration $k$, if the loss function corresponding to the quantized parameters $W^k$ violates the loss bound of Equation 2, it means that the linear estimation was inaccurate over the trust region. Thus, we reduce $\Delta_k$ in the following iteration and solve the subproblem over a smaller trust region. In this case, the calculated quantized parameters are rejected. Otherwise, we update the parameters and enlarge the trust region. But we also make sure that the trust region radius does not grow past an upper bound like $\overline{\Delta}$.

We use two measures to examine convergence of the solution: the loss function and the trust region radius. We declare convergence, if the loss function of the quantized model converges to the loss bound $\bar{\ell}$ or the trust region radius becomes very small, indicated by values $\epsilon$ and $\eta$, respectively. Note that the protocol to update the trust region radius as well as the trust region convergence condition used in this algorithm are commonly used in trust region methods.

---

**Algorithm 2** Adaptive Quantization

$k \leftarrow 0$
Initialize $\Delta_0, \overline{\Delta}$, and $\overline{\ell}$
$W^0 = Quantize(W_0, 0)$
**while** $\overline{\ell} - \mathcal{L}(W^k) \geq \epsilon$ and $\Delta_k \geq \eta$ **do**
    Define $G^k = [g_1^k ... g_n^k]^T$ for $g_i^k = |[\nabla_W \mathcal{L}(W^k)]_i|$
    Define $m_\ell^k(T) = \mathcal{L}(W^k) + G^{k^T} T$
    Find $T^k$ by solving the Trust Region Subproblem (Equation 16) with $m_\ell(T) = m_\ell^k(T)$
following section 2.3
    $\tilde{W} = Quantize(W^k, T^k)$
    **if** $\mathcal{L}(\tilde{W}) \leq \overline{\ell}$ **then**
        $W^{k+1} \leftarrow \tilde{W}$
        $\Delta_{k+1} \leftarrow min(2\Delta_k, \overline{\Delta})$
    **else**
        $\Delta_{k+1} \leftarrow \frac{1}{2}\Delta_k$
    **end if**
    $k \leftarrow k + 1$
**end while**

---

## 2.3 TRUST REGION SUBPROBLEM

In each iteration $k$, we can directly solve the subproblem of Equation 16 by writing its KKT conditions:

$$\nabla \Phi + \psi^k \nabla m_\ell^k = 0 \tag{18}$$

$$\psi^k (m_\ell^k - \overline{\ell}) = 0 \tag{19}$$

The gradient of the objective function, $\nabla \Phi = -\frac{1}{\ln 2}[\frac{1}{\tau_1} ... \frac{1}{\tau_n}]$ cannot be zero. Hence, $\psi^k > 0$ and $m_\ell^k - \overline{\ell} = 0$. We can use this to calculate $T^k$, the solution of the subproblem.

$$\nabla \Phi + \psi^k \nabla m_\ell^k = 0 \Rightarrow \forall i \in [1, n] : -\frac{1}{\tau_i^k \ln 2} + \psi^k g_i^k = 0 \Rightarrow \forall i \in [1, n] : \tau_i^k g_i^k = \frac{1}{\psi^k \ln 2} \tag{20}$$

$$\Rightarrow G^{k^T} T^k = \frac{n}{\psi^k \ln 2} \tag{21}$$

Therefore, we can write:

$$m_\ell^k - \overline{\ell} = 0 \Rightarrow \mathcal{L}(W^k) + \frac{n}{\psi^k \ln 2} - \overline{\ell} = 0 \Rightarrow \frac{n}{\psi^k \ln 2} = \frac{\overline{\ell} - \mathcal{L}(W^k)}{n} \tag{22}$$

$$\Rightarrow \forall i \in [1, n] : \tau_i^k = \frac{\overline{\ell} - \mathcal{L}(W^k)}{n g_i^k} \tag{23}$$

If the resulting tolerance vector $\|T^k\|_2 > \Delta_k$, we scale $T^k$ so that its norm would be equal to $\Delta_k$.

This solution is correct only when $g_i^k > 0$ for all $i$. It is possible, however, that there exists some $i$ for which $g_i^k = 0$. In this case, we use the following equation to calculate the solution $T^k$.

$$\tau_i^k = \begin{cases} \frac{\overline{\ell} - \mathcal{L}(W^k)}{n g_i^k} & g_i^k > 0 \\ |\omega_i^k| & g_i^k = 0 \end{cases} \tag{24}$$

We treat singular points this way for three reasons. First, a gradient of zero with respect to $\omega_i^k$ means that the loss is not sensitive to parameter values. Thus, it might be possible to eliminate it quickly. Second, this insensitivity of the loss $\mathcal{L}$ to $\omega_i^k$ means that large deviations in the parameter value would not significantly affect $\mathcal{L}$. Finally, setting the value of $\tau_i^k$ to a large value relative to other tolerances reduces their values after normalization to the trust region radius. Thus, the effect of a singularity on other parameters is reduced.

## 2.4 CHOOSING THE HYPER-PARAMETERS

The hyper-parameters, $\overline{\Delta}$, and $\overline{\ell}$, determine the speed of convergence and the final classification accuracy. Smaller trust regions result in slower convergence, while larger trust regions produce higher error rates. For $\overline{\Delta}$, since remaining values could ultimately be represented by 0 bits (pruned), we choose $2^0\sqrt{n}$. Similarly, the higher the loss bound $\overline{\ell}$, the lower the accuracy, while small values of loss bound prevent effective model size reduction. We choose this value in our algorithm on the fly. That is, we start off by conservatively choosing a small value for $\overline{\ell}$ (e.g. $\overline{\ell} = \mathcal{L}(W_0)$) and then increase it every time Algorithm 2 converges. Algorithm 3 provides the details of this process. At the beginning the loss bound is chosen to be the loss of the floating-point trained model. After quantizing the model using this bound, the bound value is increased by $Scale$. These steps are repeated $Steps$ times. In our experiments in the next section $Scale$ and $Steps$ are set to 1.1 and 20, respectively.

---

**Algorithm 3** Choosing hyper-parameters

---

Initialize $Steps$ and $Scale$
$\overline{\ell} \leftarrow \mathcal{L}(W_0)$
**while** $Steps > 0$ **do**
    Quantize model using Algorithm 2.
    $Steps \leftarrow Steps - 1$
    $\overline{\ell} \leftarrow \overline{\ell} * Scale$
**end while**

---

## 3 EVALUATION AND DISCUSSION

We evaluate adaptive quantization on three popular image classification benchmarks. For each, we first train a neural network in the floating-point domain, and then apply a pass of algorithm 3 to compress the trained model. In both these steps, we use the same batchsize to calculate the gradients and update the parameters. To further reduce the model size, we tune the accuracy of the quantized model in the floating-point domain and quantize the tuned model by reapplying a pass of algorithm 3. For each benchmark, we repeat this process three times, and experimentally show that this produces the smallest model. In the end we evaluate the accuracy and the size of the quantized models. Specifically, we determine the overall number of bits (quantization bits and the sign bits), and evaluate how much reduction in the model size has been achieved.

We note that it is also important to evaluate the potential overheads of bookkeeping for the quantization widths. However, we should keep in mind that bookkeeping has an intricate relationship with the target hardware, which may lead to radically different results on different hardware platforms. For example, our experiments show that on specialized hardware, such as the one designed by Albericio et al. (2017) for processing variable bit-width CNN, we can fully offset all bookkeeping overheads of storing quantization depths, while CPU/GPU may require up to $60\%$ additional storage. We will study this complex relationship separately, in our future work, and in the context of hardware implementation. In this paper, we limit the scope to algorithm analysis, independent of the underlying hardware architecture.

### 3.1 BENCHMARKS

We use MNIST (LeCun et al., 1998), CIFAR-10 (Krizhevsky & Hinton, 2009), and SVHN (Netzer et al., 2011) benchmarks in our experiments. MNIST is a small dataset containing $28 \times 28$ images of handwritten digits from 0 to 9. For this dataset, we use the LeNet-5 network (Lecun et al., 1998). For both CIFAR-10, a dataset of $32 \times 32$ images from 10 object classes, and SVHN, a large dataset of $32 \times 32$ real-world images of digits, we employ the same organization of convolution layers and fully connected layers as networks used in BNN (Courbariaux & Bengio, 2016). These networks are based on VGG (Simonyan & Zisserman, 2014) but have parameters in full precision. The only difference is that in the case of CIFAR-10, we use 4096 neurons instead of the 1024 neurons used in BNN, and we do not use batch normalization layers. We train these models using a Cross entropy loss function. For

CIFAR-10, we also add an $L_2$ regularization term. Table 1 shows the specifications of the baseline trained models.

Table 1: Baseline models

| Dataset | Model | Model Size | Error Rate |
|---------|-------|------------|------------|
| MNIST | LeNet-5 (Lecun et al., 1998) | $12Mb$ | 0.65% |
| CIFAR-10 | (Courbariaux & Bengio, 2016) | $612Mb$ | 9.08% |
| SVHN | (Courbariaux & Bengio, 2016) | $110Mb$ | 7.26% |

## 3.2 EXECUTION TIME

The key contributors to the computational load of the proposed technique are back propagation (to calculate gradients) and quantization (Algorithm 2). Both these operations can be completed in $\mathcal{O}(n)$ complexity. We implement the proposed quantization on Intel Core i7 CPU (3.5 GHz) with Titan X GPU performing training and quantization. The timing results of the algorithm have been summarized in Table 2.

Table 2: Timing results for training and quantization of the benchmark models in **seconds**

| Dataset | Training | Quantization |
|---------|----------|--------------|
| MNIST | 120 | 300 |
| CIFAR-10 | 4560 | 4320 |
| SVHN | 1920 | 2580 |

## 3.3 QUANTIZATION RESULTS

We evaluate the performance of the quantization algorithm by analyzing the compression rate of the models and their respective error rates after each pass of quantization excluding retraining. As discussed in section 2, the quantization algorithm will try to reduce parameter precisions while maintaining a minimum classification accuracy. Here we present the results of these experiments.

Figure 1a depicts the error rate of LeNet trained on the MNIST dataset as it is compressed through three passes of adaptive quantization and retraining. As this figure shows, the second and third passes tend to have smaller error rates compared to the first pass. But, at the end of the second pass the error rate is higher than the first one. This can be improved by increasing the number of epochs for retraining or choosing a lower cost bound ($\bar{\bar{\ell}}$). By the end of the third pass, the highest compression rate is achieved. However, due to its small difference in compression rate compared to its preceding pass, we do not expect to achieve more improvements by continuing the retraining process.

We also evaluate the convergence speed of the algorithm by measuring the compression rate ($1 - \frac{size\ of\ quantized\ model}{size\ of\ original\ model}$) of the model after each iteration of the loop in Algorithm 2. The results of this experiment for one pass of quantization of Algorithm 3 have been depicted in Figure 1b. As shown in the figure, the size of the model is reduced quickly during the initial iterations. This portion of the experiment corresponds to the part of Figure 1a where the error rate is constant. However, as quantization continues, the model experiences diminishing returns in size reduction. After 25 iterations, little reduction in model size is achieved. The lower density of data points, past this point, is due to the generated steps failing the test on the upper bound on the loss function in Algorithm 2. Consequently, the algorithm reduces the trust region size and recalculates the tolerance steps.

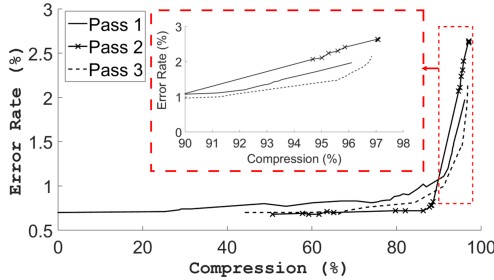 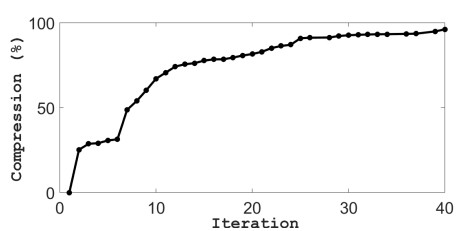

(a) Error rate increases as the model is compressed when retraining is applied

(b) Convergence of quantization. Iterations refers to iterations of the loop in algorithm 2.

Figure 1: Quantization of LeNet-5 model trained on MNIST dataset

## 4 COMPARISON WITH RELATED APPROACHES

The generality of the proposed technique makes it adaptable to other model compression techniques. In this section, we review some of the most notable of these techniques and examine the relationship between their approaches and ours. Specifically, we will explain how the proposed approach subsumes previous ones and how it can be specialized to implement them in an improved way.

**Pruning:** In pruning, small model parameters in a trained network are set to zero, which means that their corresponding connections are eliminated from the network. Han et al. (2015) showed that by using pruning, the number of connections in the network can be reduced substantially. Adaptive Quantization confirms similar observations, that in a DNN most parameters can be eliminated. In fact, Adaptive Quantization eliminates $5.6\times$, $5.6\times$, and $5.3\times$ of the parameters in the networks trained for MNIST, CFAR, and SVHN, respectively. These elimination rates are lower compared to their corresponding values achieved by Deep Compression. The reason for this difference is that Deep Compression amortizes for the loss of accuracy due to pruning of a large number of parameters by using full precision in the remaining ones. In contrast, Adaptive Quantization eliminates fewer parameters and instead quantizes the remaining ones.

While Adaptive Quantization identifies connections that can be eliminated automatically, Deep Compression identifies them by setting parameters below a certain threshold to zero. However, this technique might not be suitable in some cases. For an example, we consider the network model trained for CIFAR-10. This network has been trained using $L_2$ regularization, which helps avoid overfitting. As a result, the trained model has a large population of small-value parameters. Looking at Figure 2a, which shows the distribution of parameter values in the CIFAR-10 model, we see that most populate a small range around zero. Such cases can make choosing a good threshold for pruning difficult.

**Weight Sharing:** The goal of weight-sharing is to create a small dictionary of parameters (weights). The parameters are grouped into bins whose members will all have the same value. Therefore, instead of storing parameters themselves, we can replace them with their indexes in the dictionary. Deep Compression (Han et al., 2015) implements weight-sharing by applying k-means clustering to trained network parameters. Similarly, Samragh et al. (2017) implement weight sharing by iteratively applying k-means clustering and retraining in order to find the best dictionary . In both of these works, the number of dictionary entries are fixed in advance. Adaptive Quantization, on the other hand, produces the bins as a byproduct and does not make assumptions on the total number of bins.

Deep Compression also identifies that the accuracy of the network is less sensitive to the fully connected layers compared to the convolution layers. Because of that, it allocates a smaller dictionary for storing them. Looking at Figure 2b and Figure 2c, we can see that results of Adaptive Quantization are consistent with this observation. These figures show the distribution of quantization widths for one convolution layer and one fully connected layer in the quantized CIFAR-10 network. It is clear from these figures that parameters in the fully connected layer, on average require smaller quantization widths.

**Binarization and Quantization:** Binarized Neural Networks (Courbariaux et al., 2015; Courbariaux & Bengio, 2016) and Quantized Neural Networks (Hubara et al., 2016) can reduce model size by

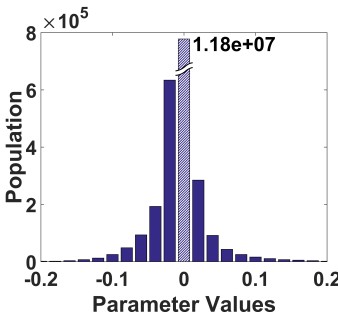 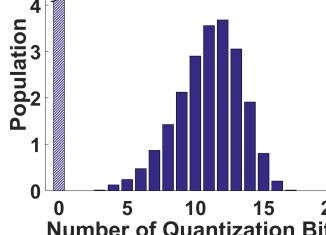 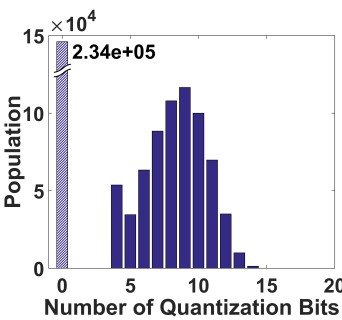

| (a) Distribution of parameter values after quantization. | (b) Distribution of quantization widths in the first convolution layer. | (c) Distribution of quantization widths in the first fully connected layer. |

Figure 2: Distributions of parameter values and quantization widths for the CIFAR-10 network. Pruned parameters have been depicted using a hatch pattern and their populations have been reported besides their corresponding bars.

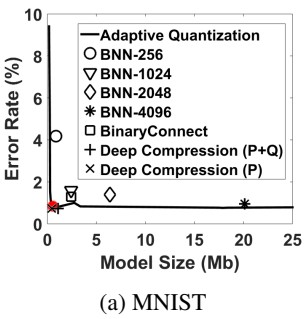 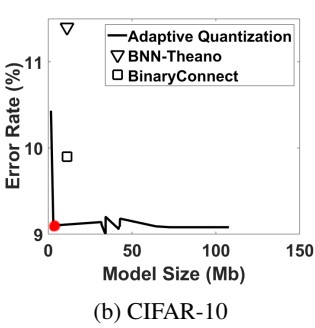 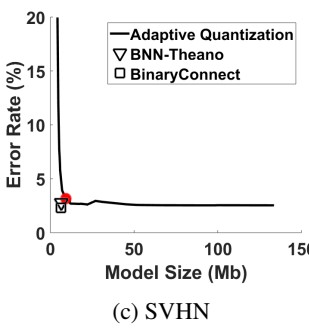

| (a) MNIST | (b) CIFAR-10 | (c) SVHN |

Figure 3: Trade-off between accuracy and error rate for benchmark datasets. The optimal points for MNIST, CIFAR-10, and SVHN (highlighted red) achieve $64\times$, $35\times$, and $14\times$ compression and correspond to 0.12%, -0.02%, and 0.7% decrease in accuracy, respectively. BNN-2048, BNN-24096, BNN-Theano are from (Courbariaux & Bengio, 2016); BinaryConnect is from (Courbariaux et al., 2015); and Deep Compression is from (Han et al., 2015)

assuming the same quantization precision for all parameters in the network or all parameters in a layer. In the extreme case of BNNs, parameters are quantized to only 1 bit. In contrast, our experiments show that through pruning and quantization, the proposed approach quantizes parameters of the networks for MNIST, CIFAR-10, and SVHN by equivalent of 0.03, 0.27, and 1.3 bits per parameter with 0.12%, $-0.02\%$, and 0.7% decrease in accuracy, respectively. Thus, our approach produces competitive quantizations. Furthermore, previous quantization techniques often design quantized training algorithms to maintain the same parameter precisions throughout training. These techniques can be slower than full-precision training. In contrast, Adaptive Quantization allows for unique quantization precisions for *each parameter*. This way, for a trained model, it can find a notably smaller network, indicating the limits of quantization. In addition, it does not require quantized training.

Next, we compare Adaptive Quantization with previous works on compressing neural network models, in reducing the model size. The results for these comparisons have been presented in Figure 3, which shows the trade-offs that Adaptive Quantization offers between accuracy and model size. As this figure shows, Adaptive Quantization in many cases outperforms previous methods and consistently produces compressions better than or comparable to state-of-the-art. In particular, we mark an optimal trade-off for each model in Figure 3 (red highlight). In these points, the proposed method achieves $64\times$, $35\times$, and $14\times$ compression and correspond to 0.12%, -0.02%, and 0.7% decrease in accuracy, respectively. This always improves or is comparable to the state-of-the-art of Quantization (BNN and BinaryConnect) and Pruning and Weight-Sharing (Deep Compression). Below, we discuss these trade-offs in more details.

In both MNIST and CIFAR-10, Adaptive Quantization produces curves below the results achieved in BNN (Courbariaux & Bengio, 2016) and BinaryConnect (Courbariaux et al., 2015), and it shows a smaller model size while maintaining the same error rate or, equivalently, the same model size with a smaller error rate. In the case of SVHN, BNN achieves a slightly better result compared to Adaptive Quantization. This is due in part to the initial error rate of our full-precision model being higher than the BNN model.

Comparison of Adaptive Quantization against Deep Compression (when pruning is applied) for MNIST shows some similarity between the two techniques, although deep compression achieves a slightly smaller error rate for the same model size. This difference stems from the different approaches of the two techniques. First, Deep Compression uses full-precision, floating-point parameters while Adaptive Quantization uses fixed-point quantized parameters which reduce the computational complexity (fixed-point operation vs. floating-point operation). Second, after pruning the network, Deep Compression performs a complete retraining of the network. In contrast, Adaptive Quantization performs little retraining.

Furthermore, Adaptive Quantization can be used to identify groups of parameters that need to be represented in high-precision formats. In a more general sense, the flexibility of Adaptive Quantization allows it to be specialized for more constrained forms of quantization. For example, if the quantization requires that all parameters in the same layer have the same quantization width, Adaptive Quantization could find the best model. This could be implemented by solving the same minimization problem as in Equation 16, except the length of $T$ would be equal to the number of layers. In such a case, $G^k$ in Algorithm 2 will be defined as $G^k = [g_1^k...g_m^k]^T$ assuming a total of $m$ layers where $g_i^k$ would be $\sum_{j \in J_i} |\frac{\partial \mathcal{L}}{\partial \omega_j}(W^k)|$, and $J_i$ the set of parameter indexes belonging to layer $i$. Then, each of the resulting $m$ tolerance values of the subproblem of section 2.3 will be applied to all parameters of one layer. The rest of the solution would be the same.

## 5 CONCLUSION

In this work, we quantize neural network models such that only parameters critical to the accuracy are represented with high precision. The goal is to minimize data movement and simplify computations needed for inference in order to accelerate implementations on resource constrained hardware. To achieve acceleration, the proposed technique prunes unnecessary parameters or reduces their precisions. Combined with existing fixed-point computation techniques such as SWAR (Cameron & Lin, 2009) or Bit-pragmatic computation (Albericio et al., 2017), we expect these small fixed-point models to achieve fast inference with high accuracies. We have confirmed the effectiveness of this technique through experiments on several benchmarks. Through this technique, our experiments show, DNN model sizes can be reduced significantly without loss of accuracy. The resulting models are significantly smaller than state-of-the-art quantization technique. Furthermore, the proposed Adaptive Quantization can provide similar results to floating-point model compression techniques.

## 6 ACKNOWLEDGMENTS

We would like to express our gratitude to Professor Stephen J. Wright, Professor Jerry Zhu, and Professor Dimitris Papailiopoulos from UW-Madison for their helpful discussions throughout the process of writing this paper. We also want to thank Yue Zha for helping improve the writing quality of the manuscript.

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
