# OpenReview forum: "Adaptive Quantization of Neural Networks"
_ICLR.cc/2018/Conference — Accept (Poster)_

### Official Review · AnonReviewer3 · 2017-11-27
**Interesting intuitive idea; evaluation needs more clarification**

**Rating:** 6
**Confidence:** 3

**Review:**

The authors present an interesting idea to reduce the size of neural networks via adaptive compression, allowing the network to use high precision where it is crucial and low precision in other parts. The problem and the proposed solution is well motivated. However, there are some elements of the manuscript that are hard to follow and need further clarification/information. These need to definitely be addressed before this paper can be accepted.

Specific comments/questions:
- Page 1: Towards the bottom, in the 3rd to last line, reference is missing.
- Page 1: It is a little hard to follow the motivation against existing methods.
- Page 2: DenseNets and DeepCompression need citations
- Lemma 2.1 seems interesting - is this original work? This needs to be clarified.
- Lemma 2.2: Reference to Equation 17 (which has not been presented in the manuscript at this point) seems a little confusing and I am unable to following the reasoning and the subsequent proof which again refers to Equation 17.
- Alg 2: Should it be $\Delta$ or $\Delta_{k+1}$? Because in one if branch, we use $\Delta$, in the other, we use the subscripted one.
- Derivation in section 2.3 has some typographical errors.
- What is $d$ in Equation 20 (with cases)? Without this information, it is unclear how the singular points are handled.
- Page 6, first paragraph of Section 3: The evaluation is a little confusing - when is the compression being applied during the training process, and how is the training continued post-compression? What does each compression 'pass' constitute of?
- Figure 1b: what is the 'iteration' on the horizontal axis, is it the number of iterations of Alg3 or Alg2? Hoping it is Alg3 but needs to be clarified in the text.
- Section 3: What about compression results for CIFAR and SVNH?

---

> ### Author Response · Authors · 2018-01-05
> **We have clarified the evaluation methodology in the updated version of the paper**
>
> We appreciate your insightful comments. In the updated version of the paper, we have fixed the missing references and typos, and clarified the evaluation methodology as well as the other points mentioned by the reviewer. The changes have been highlighted in blue. Here, we address specific questions by the reviewer one-by-one.
>
> 1. Page 1: Towards the bottom, in the 3rd to last line, reference is missing.
>
> - Added references Hubara (2016a) and Han (2015).
>
>
> 2. Page 1: It is a little hard to follow the motivation against existing methods.
>
> - Modified the discussion in the introduction (highlighted blue).
>
>
> 3. Page 2: DenseNets and DeepCompression need citations
>
> - Added references Huang (2017) and Han (2015) in section 1.
>
>
> 4. Lemma 2.1 seems interesting - is this original work? This needs to be clarified.
>
> - Lemma 2.1 is an original contribution of the paper. We added clarification in Section 2.
>
>
> 5. Lemma 2.2: Reference to Equation 17 (which has not been presented in the manuscript at this point) seems a little confusing and I am unable to following the reasoning and the subsequent proof which again refers to Equation 17.
>
> - We revised the cross references in the proof of lemma 2.2. The constraint refers to the definitions in equations  11 and 12.
>
>
> 6. Alg 2: Should it be $\Delta$ or $\Delta_{k+1}$? Because in one if branch, we use $\Delta$, in the other, we use the subscripted one.
>
> - Added the subscript in algorithm 2.
>
>
> 7. Derivation in section 2.3 has some typographical errors.
>
> - Fixed the typographical errors.
>
>
> 8. What is $d$ in Equation 20 (with cases)? Without this information, it is unclear how the singular points are handled.
>
> - $d$ in equation 20 refers to the difference between the loss bound $\overline{l}$ and the loss in the current iteration of the algorithm $l(W_k)$: $d = \overline{l}-l(W_k)$. We have modified equation 20 accordingly.
>
>
> 9. Page 6, first paragraph of Section 3: The evaluation is a little confusing
>
> - Revised the first paragraph of section 3 to clarify the process of evaluation (highlighted blue).
>
>
> 10. when is the compression being applied during the training process, and how is the training continued post-compression? What does each compression 'pass' constitute of?
>
> - We added additional explanation in section 3 regarding when compression is performed and what a pass of compression constitutes. Specifically, adaptive quantization is applied to a model after the training is complete. The retraining steps after the compression are performed  in full-precision, floating-point domain. Also, each pass of compression refers to a complete execution of algorithm 3.
>
>
> 11. Figure 1b: what is the 'iteration' on the horizontal axis, is it the number of iterations of Alg3 or Alg2? Hoping it is Alg3 but needs to be clarified in the text.
>
> - We clarified the definition of iteration in figure 1. Each iteration, refers to one iteration of the loop in algorithm 2.
>
>
> 12. Section 3: What about compression results for CIFAR and SVNH?
>
> - We have added the compression results for the optimal trade-off for all three datasets in the revised version (Figure 3). We have further added comparison with BinaryConnect for all datasets and shown that the original conclusions hold. That is, the proposed algorithm almost always outperforms state-of-the-art of quantization (BinaryConnect and BNN) and consistently produces competitive results.

---

### Official Review · AnonReviewer2 · 2017-11-28
**The paper proposes a method for quantizing neural networks that allows weights to be quantized with different precision depending on their importance**

**Rating:** 6
**Confidence:** 4

**Review:**

I have read the responses to the concerns raised by all reviewers. I find the clarifications and modifications satisfying, therefore I keep my rating of the paper to above acceptance threshold.

-----------------
ORIGINAL REVIEW:

The paper proposes a method for quantizing neural networks that allows weights to be quantized with different precision depending on their importance, taking into account the loss. If the weights are very relevant, it assigns more bits to them, and in the other extreme it does pruning of the weights.

This paper addresses a very relevant topic, because in limited resources there is a constrain in memory and computational power, which can be tackled by quantizing the weights of the network. The idea presented is an interesting extension to weight pruning with a close form approximate solution for computing the adaptive quantization of the weights.

The results presented in the experimental section are promising. The quantization is quite cheap to compute and the results are similar to other state-of-the-art quantization methods.
From the tables and figures, it is difficult to grasp the decrease in accuracy when using the quantized model, compared to the full precision model, and also the relative memory compression. It would be nice to have this reference in the plots of figure 3.  Also, it is difficult to see the benefits in terms of memory/accuracy compromise since not all competing quantization techniques are compared for all the datasets.
Another observation is that it seems from figure 2 that a lot of the weights are quantized with around 10 bits, and it is not clear how the compromise accuracy/memory can be turned to less memory, if possible. It would be interesting to know an analogy, for instance, saying that this adaptive compression in memory would be equivalent to quantizing all weights with n bits.

OTHER COMMENTS:

-missing references in several points of the paper. For instance, in the second paragraph of the introduction, 1st paragraph of section 2.

- few typos:
*psi -> \psi in section 2.3
*simply -> simplify in proof of lemma 2.2
*Delta -> \Delta in last paragraph of section 2.2
*l2 -> L_2 or l_2 in section 3.1 last paragraph.

---

> ### Author Response · Authors · 2018-01-05
> **Modified the paper as per the reviewer's suggestions to add more discussion regarding the figures**
>
> Thank you for your valuable comments. We have modified the paper accordingly and highlighted the changes in blue. We have also resolved the missing references and typos. Below, we discuss the points mentioned by the reviewer in detail.
>
> Question: From the tables and figures, it is difficult to grasp the decrease in accuracy when using the quantized model, compared to the full precision model, and also the relative memory compression. It would be nice to have this reference in the plots of figure 3.
>
> Answer: Thanks for pointing this out. In the revised version, we highlight the optimal trade-off between accuracy and model size for each model in Figure 3. We further report the accuracy and the reduction in the model size for these optimal models. We observe compression ratios of these optimal models equal to 64x, 35x, and 13x (corresponding to 98.4%, 97%, and 92% reductions in model size) for MNIST, CIFAR-10, and SVHN, with 0.12%, -0.02%, and 0.7% decrease in accuracy, respectively. We modified section 4 to clarify these results.
>
>
>
> Question: Also, it is difficult to see the benefits in terms of memory/accuracy compromise since not all competing quantization techniques are compared for all the datasets.
>
> Answer: In Figure 3, we have added comparisons with the BinaryConnect technique for all three datasets. This technique can often improve the accuracy of BNN with the same model size. Yet, these comparisons confirm our original results. That is, the proposed method almost always outperforms state-of-the-art of quantization (BinaryConnect and BNN) and consistently produces competitive results. We have modified section 4 with the discussion of these results.
>
>
>
> Question: Another observation is that it seems from figure 2 that a lot of the weights are quantized with around 10 bits, and it is not clear how the compromise accuracy/memory can be turned to less memory, if possible. It would be interesting to know an analogy, for instance, saying that this adaptive compression in memory would be equivalent to quantizing all weights with n bits.
>
> Answer: Figures 2 (a, b, c), for clarity, only show non-pruned parameters, which comprise a small portion of the original parameters of the model. Taking these parameters into account, adaptive quantization compresses MNIST, CIFAR-10, and SVHN models to equivalent of 0.03, 0.27, and 1.3 bits per parameter, respectively (results are for the optimal trade-off points highlighted in Figure 3). These are all significantly smaller or  comparable to state-of-the-art of quantization, that is, BNN and BinaryConnect (1 bit per parameter). We have modified section 4 with these clarifications and updated Figure 2 (a, b, c) to include the pruned parameters for comparison with non-pruned parameters.
>
>
>
> Question: missing references in several points of the paper. For instance, in the second paragraph of the introduction, 1st paragraph of section 2.
>
> Answer: Thanks. We have included the references in the revised version of the paper.

---

### Official Review · AnonReviewer1 · 2017-12-04
**Clarity issues; concerns about practical relevance**

**Rating:** 6
**Confidence:** 4

**Review:**

Revised Review:

The authors have addressed most of my concerns with the revised manuscript. I now think the paper does just enough to warrant acceptance, although I remain a bit concerned that since the benefits are only achievable with customized hardware, the relevance/applicability of the work is somewhat limited.

Original Review:

The paper proposes a technique for quantizing the weights of a neural network, with bit-depth/precision varying on a per-parameter basis. The main idea is to minimize the number of bits used in the quantization while constraining the loss to remain below a specified upper bound. This is achieved by formulating an upper bound on the number of bits used via a set of "tolerances"; this upper bound is then minimized while estimating any increase in loss using a first order Taylor approximation.

I have a number of questions and concerns about the proposed approach. First, at a high level, there are many details that aren't clear from the text. Quantization has some bookkeeping associated with it: In a per-parameter quantization setup it will be necessary to store not just the quantized parameter, but also the number of bits used in the quantization (takes e.g. 4-5 extra bits), and there will be some metadata necessary to encode how the quantized value should be converted back to floating point (e.g., for 8-bit quantization of a layer of weights, usually the min and max are stored). From Algorithm 1 it appears the quantization assumes parameters in the range [0, 1]. Don't negative values require another bit? What happens to values larger than 1? How are even bit depths and associated asymmetries w.r.t. 0 handled (e.g., three bits can represent -1, 0, and 1, but 4 must choose to either not represent 0 or drop e.g. -1)? None of these details are clearly discussed in the paper, and it's not at all clear that the estimates of compression are correct if these bookkeeping matters aren't taken into account properly.

Additionally the paper implies that this style of quantization has benefits for compute in addition to memory savings. This is highly dubious, since the method will require converting all parameters to a standard bit-depth on the fly (probably back to floating point, since some parameters may have been quantized with bit depth up to 32). Alternatively custom GEMM/conv routines would be required which are impossible to make efficient for weights with varying bit depths. So there are likely not runtime compute or memory savings from such an approach.

I have a few other specific questions: Are the gradients used to compute \mu computed on the whole dataset or minibatches? How would this scale to larger datasets? I am confused by the equality in Equation 8: What happens for values shared by many different quantization bit depths (e.g., representing 0 presumably requires 1 bit, but may be associated with a much finer tolerance)? Should "minimization in equation 4" refer to equation 3?

In the end, while do like the general idea of utilizing the gradient to identify how sensitive the model might be to quantization of various parameters, there are significant clarity issues in the paper, I am a bit uneasy about some of the compression results claimed without clearer description of the bookkeeping, and I don't believe an approach of this kind has any significant practical relevance for saving runtime memory or compute resources.

---

> ### Author Response · Authors · 2018-01-05
> **We clarify the points mentioned by the reviewer and address the practicality concerns**
>
> Thank you for your insightful comments. We have modified the manuscript based on the questions from the reviewer. The changes and additions to the paper have been highlighted in blue. Below we discuss each of the questions in more detail one-by-one.
>
> Question: I have a number of questions and concerns about the proposed approach. First, at a high level, there are many details that aren't clear from the text. Quantization has some bookkeeping associated with it: In a per-parameter quantization setup it will be necessary to store not just the quantized parameter, but also the number of bits used in the quantization (takes e.g. 4-5 extra bits), and there will be some metadata necessary to encode how the quantized value should be converted back to floating point (e.g., for 8-bit quantization of a layer of weights, usually the min and max are stored). From Algorithm 1 it appears the quantization assumes parameters in the range [0, 1]. Don't negative values require another bit? What happens to values larger than 1? How are even bit depths and associated asymmetries w.r.t. 0 handled (e.g., three bits can represent -1, 0, and 1, but 4 must choose to either not represent 0 or drop e.g. -1)? None of these details are clearly discussed in the paper, and it's not at all clear that the estimates of compression are correct if these bookkeeping matters aren't taken into account properly.
>
> Answer: We agree with the reviewer that it is important to evaluate the potential overhead of bookkeeping. However, we should also have in mind that bookkeeping has an intricate relationship with the target hardware, which may lead to radically different results on different hardware platforms (ranging from 0 to ~60%). For example, our experiments show that on specialized hardware, such as the one designed by Albericio et al (2017) for processing variable bit width CNN, we can fully offset all bookkeeping overheads of storing quantization depths, while CPU/GPU may require up to 60% additional storage. We will study this complex relationship separately, in our future work, and in the context of hardware implementation. In this paper, we limit the scope to algorithm analysis, independent of underlying hardware architectures. We note that in this analysis, we have evaluated the metadata as well as the additional sign bits. The metadata overhead is negligible (about 4 bytes per layer) due to the balanced quantization of algorithm 1 which divides the range [0,1] into equally sized partitions and assigns a single bit to each parameter. As we discuss in the answer to the next question, this scheme eliminates the need to convert parameters back to floating-point, and computations can be performed directly on the quantized values. For example, the 5-bit signed value 01011, for example, represents 2^(-1)+2^(-3)+2^(-4)=0.6875 (the initial 0 bit represents a positive value), which can be easily multiplied with other values using fixed-point shifts and additions. If it is necessary to have parameters in a larger range, say [-S, S], a scale value like S (4 bytes of metadata) could be allocated for each layer, that is applied to the output of that layer. We have clarified these points in the updated version of the paper, in section 2 and section 3.
>
> Albericio, Jorge, et al. "Bit-pragmatic deep neural network computing." Proceedings of the 50th Annual IEEE/ACM International Symposium on Microarchitecture. ACM, 2017.
>
>
>
> Question: Additionally the paper implies that this style of quantization has benefits for compute in addition to memory savings. This is highly dubious, since the method will require converting all parameters to a standard bit-depth on the fly (probably back to floating point, since some parameters may have been quantized with bit depth up to 32). Alternatively custom GEMM/conv routines would be required which are impossible to make efficient for weights with varying bit depths. So there are likely not runtime compute or memory savings from such an approach.
>
> Answer: We agree that on CPU/GPU interpreting variable-bit width parameters may incur computational costs. However, our quantization scheme significantly reduces the necessary computation on our target platforms, that is, specialized hardware like Alberricio et al (2017) or configurable hardware like FPGAs. These platforms can directly process the variable-bit width, fixed-point parameters without the need to convert them into floating point, and can implement custom computation units to efficiently perform matrix multiplication/convolutions by taking advantage of the small quantization depths of the parameters in the quantized model. We note that in our experiments, parameters are often quantized with far fewer bits than 32, with little to no accuracy loss. Thus, our approach can significantly accelerate performance on this class of hardware by minimizing the required computations. We have clarified this in section 2.

---

> > ### Author Response · Authors · 2018-01-05
> > **Other specific questions**
> >
> >
> > Question: I have a few other specific questions: Are the gradients used to compute \mu computed on the whole dataset or minibatches? How would this scale to larger datasets?
> >
> > Answer: Gradients are calculated on minibatches. As we have specified in section 3, we use the same batch size for training and quantization to keep the computation time short. Our experiments show that this decision does not have a negative effect on the accuracy of the quantized model. Thus, as long as we choose representative batch sizes, as we do for training, the algorithm scales to larger datasets with no need for modifications. We have modified Section 3 for clarification.
> >
> >
> >
> > Question: I am confused by the equality in Equation 8: What happens for values shared by many different quantization bit depths (e.g., representing 0 presumably requires 1 bit, but may be associated with a much finer tolerance)?
> >
> > Answer: This equation explores the worst case for quantization error and shows that in this case the quantization depth is bounded by negative logarithm of the tolerance. In general, we can expect the quantization depth to be smaller than this value. That is because Algorithm 1 minimizes the bit width of a parameter with respect to its tolerance. If multiple bit widths satisfy this requirement, the smallest is always chosen. For example, a parameter with the signed value equal to 0.25 can be represented by both 001 and 0010. Algorithm 1 however, will always return the former. We have modified Section 2 for clarification.
> >
> >
> >
> > Question: Should "minimization in equation 4" refer to equation 3?
> >
> > Answer: Yes. Thank you for pointing this out. We have corrected the typo.

---

### Public Comment · ~Sriharsha_Chandra_Mohan1 · 2019-06-19
**Some doubts in the paper**


1) How is weight sharing implemented here? How are the bins produced as a by-product and what is their significance in adaptive quantization?
2) When and how much retraining is done exactly? Can we achieve better accuracy by performing a full retraining on the quantized model?

PS*  Is there a code available for this?

---

### Decision · Program_Chairs · 2018-01-29
**ICLR 2018 Conference Acceptance Decision**

**Decision:**

Accept (Poster)

**Comment:**

Given the changes to the paper, the reviewers agree that the paper meets the bar for publication at ICLR. There are some concerns regarding the practical impact on CPUs and GPUs. I ask the authors to clearly discuss the impact on different hardware. One can argue if adaptive quantization techniques are helpful, then there is a chance that future hardware will support them. All of the experiments are conducted on toy datasets. Please consider including some experiments on Imagenet as well.